# Longitudinal dynamics of SARS-CoV-2-specific cellular and humoral immunity after natural infection or BNT162b2 vaccination

Patricia Almendro-Vázquez[1], Rocio Laguna-Goya[1,2,3]*, Maria Ruiz-Ruigomez[1,4], Alberto Utrero-Rico[1], Antonio Lalueza[1,3,4,5], Guillermo Maestro de la Calle[1,4], Pilar Delgado[6], Luis Perez-Ordoño[7], Eva Muro[7], Juan Vila[7], Isabel Zamarron[7], Miguel Moreno-Batanero[1], Marta Chivite-Lacaba[1], Francisco Javier Gil-Etayo[1,2], Carmen Martín-Higuera[1,8], María Ángeles Meléndez-Carmona[1,8], Carlos Lumbreras[1,3,4,5], Irene Arellano[6], Balbino Alarcon[6], Luis Miguel Allende[1,2,9], Jose Maria Aguado[1,4,10], Estela Paz-Artal[1,2,3,9]

1 Instituto de Investigación Sanitaria Hospital 12 de Octubre (imas12), Madrid, Spain, 2 Department of Immunology, Hospital Universitario 12 de Octubre, Madrid, Spain, 3 Centro de Investigación Biomédica en Red de Enfermedades Infecciosas (CIBER), Instituto de Salud Carlos III, Madrid, Spain, 4 Department of Internal Medicine, Hospital Universitario 12 de Octubre, Madrid, Spain, 5 Department of Medicine, School of Medicine, Universidad Complutense de Madrid, Madrid, Spain, 6 Centro de Biologia Molecular Severo Ochoa, Consejo Superior de Investigaciones Cientificas (CSIC), Universidad Autonoma de Madrid, Madrid, Spain, 7 Department of Emergency Medicine, Hospital Universitario 12 de Octubre, Madrid, Spain, 8 Department of Clinical Microbiology, Hospital Universitario 12 de Octubre, Madrid, Spain, 9 Department of Immunology, Ophthalmology and ENT, Universidad Complutense de Madrid, Madrid, Spain, 10 Unit of Infectious Diseases, Hospital Universitario 12 de Octubre, Madrid, Spain

* rocio.laguna@salud.madrid.org

## Abstract

The timing of the development of specific adaptive immunity after natural SARS-CoV-2 infection, and its relevance in clinical outcome, has not been characterized in depth. Description of the long-term maintenance of both cellular and humoral responses elicited by real-world anti-SARS-CoV-2 vaccination is still scarce. Here we aimed to understand the development of optimal protective responses after SARS-CoV-2 infection and vaccination. We performed an early, longitudinal study of S1-, M- and N-specific IFN-γ and IL-2 T cell immunity and anti-S total and neutralizing antibodies in 88 mild, moderate or severe acute COVID-19 patients. Moreover, SARS-CoV-2-specific adaptive immunity was also analysed in 234 COVID-19 recovered subjects, 28 uninfected BNT162b2-vaccinees and 30 uninfected healthy controls. Upon natural infection, cellular and humoral responses were early and coordinated in mild patients, while weak and inconsistent in severe patients. The S1-specific cellular response measured at hospital arrival was an independent predictive factor against severity. In COVID-19 recovered patients, four to seven months post-infection, cellular immunity was maintained but antibodies and neutralization capacity declined. Finally, a robust Th1-driven immune response was developed in uninfected BNT162b2-vaccinees. Three months post-vaccination, the cellular response was comparable, while the humoral response was consistently stronger, to that measured in COVID-19 recovered patients. Thus, measurement of both humoral and cellular responses provides information on

**Data Availability Statement:** All relevant data are within the manuscript and its Supporting Information files.

**Funding:** This study was supported by the Instituto de Salud Carlos III, Spanish Ministry of Science and Innovation (COVID-19 research call COV20/00181) — co-financed by the European Development Regional Fund "A way to achieve Europe", Operative Program Intelligent Growth 2014-2020, by Consejeria de Sanidad de la Comunidad de Madrid (CIVICO study 2020/0082) and by a private donation (code 2021/55). RLG holds a research contract "Rio Hortega" (CM19/00120) from the Instituto de Salud Carlos III, Spanish Ministry of Science and Innovation. MCL holds a predoctoral fellowship (FPU19/06393) from the Spanish Ministry of Science and Innovation. The funders had no role in study design, data collection and analysis, decision to publish, or preparation of the manuscript.

**Competing interests:** The authors have declared that no competing interests exist.

prognosis and protection from infection, which may add value for individual and public health recommendations.

## Author summary

In this work we describe the prognostic value of early detection of SARS-CoV-2-specific T cell response in acute COVID-19, as patients without a prompt SARS-CoV-2-specific T cell response progress to severe COVID-19. We also show that the presence of specific T cells against SARS-CoV-2 when patients arrive to the emergency room is a protective factor against developing severe COVID-19, independently of the age and gender of the patient, which are two major known contributors to disease outcome. In addition, we show robust cellular and humoral immune responses persist 3 months after real-world vaccination.

## Introduction

SARS-CoV-2 infection is responsible for the current coronavirus disease 2019 (COVID-19) pandemic. Most SARS-CoV-2-infected subjects show mild or no symptoms, however, an important proportion of patients require hospitalization and some of them die [1]. This disparity in disease severity has been associated with characteristics such as age, male sex and comorbidities [1–3]. The immune response elicited by SARS-CoV-2 in each individual could also be related to the clinical outcome of the infection. An exaggerated innate immune activation has been extensively described in patients with a severe course of the disease [4–8], while the role of adaptive immune response in protection from this infection is less understood.

SARS-CoV-2-specific T cells have been detected during convalescence and up to eight months post-symptom onset (PSO) in COVID-19 patients [9–14]. However, the development of the specific cellular immune response during the acute phase of infection has not yet been characterized in-depth, except for several studies, with a limited number of patients, indicating that early induction of SARS-CoV-2-specific T cells can limit COVID-19 severity [15–17]. Humoral responses to SARS-CoV-2 have been described from acute infection to convalescence and up to 8 months post-infection. Levels of specific antibodies have been positively correlated with disease severity [11,18,19] and a delayed production of neutralizing antibodies has been associated with fatal COVID-19 [20]. The correct kinetics and balance between the two arms of the adaptive immune response may be crucial for the resolution of the infection.

The mRNA vaccine BNT162b2 was approved after showing safety and a 95% efficacy in preventing COVID-19 [21]. A two-dose regimen, 21 days apart, was established with the previous vaccine candidate BNT162b1 in order to develop neutralizing antibodies [22]. BNT162b2 has proven its effectiveness in reducing symptomatic SARS-CoV-2 infection [23,24]. There are studies on the immunogenicity of BNT162b2 which expand up to a month post-vaccination [25–27] and on the maintenance of vaccine-induced humoral responses six months post-vaccination [28,29], however, data on the dynamics and long-term duration of SARS-CoV-2 specific T cells after real-world vaccination are still lacking.

Here we longitudinally characterize the adaptive immune responses during acute and convalescent COVID-19 phases in three well-defined patient cohorts of different severity, in recovered patients up to 7 months PSO and in uninfected subjects up to three months after full vaccination. Our acute infection cohort represents the largest of its kind, with 88 patients

recruited before any medication was administered (in particular no immunomodulatory drugs) and has allowed us to statistically demonstrate that the presence of specific T cells against SARS-CoV-2 when patients arrive to the emergency room is a protective factor against developing severe COVID-19, independently of the age and gender of the patient. We also describe a correlation between higher numbers of specific T cells and lower viral load at diagnosis, which could explain the protection conferred by T cells. In addition, we characterize the vaccine-elicited adaptive immunity at five time-points and show robust cellular and humoral immune responses persist 3 months after real-world vaccination.

## Results

### Study design and participants

Cellular and humoral SARS-CoV-2-specific immune responses were analyzed in patients with natural infection, uninfected healthy controls and BNT162b2 vaccinated individuals. Samples were obtained upon arrival to the emergency room (ER), prior to any treatment administration, in patients with acute COVID-19 who were later classified according to their maximum disease severity into mild (N = 32), moderate (N = 34) and severe (N = 22) (see classification criteria in Fig 1A). Mild patients were not hospitalized, and moderate and severe patients required hospitalization. Clinical and demographic characteristics, WHO ordinal scale [30] and treatments are detailed in Table 1. Patients attended the ER a week PSO on average in the 3 groups. In moderate and severe patients, a follow-up sample was taken a week after hospitalization. In mild and moderate patients another sample was taken during the convalescent phase, a month PSO.

Samples from 234 COVID-19 recovered patients were obtained, between 4 and 7 months PSO. All recovered patients had been hospitalized during the acute phase with moderate or severe COVID-19 (WHO ordinal scale of 3–7). The use of corticosteroids in the recovered cohort was lower than in the acute infection cohort, consistent with a change in treatment guidelines from the first to the second pandemic wave (Table 1).

A 30 healthy control cohort included subjects with SARS-CoV-2 negative serology and no prior known contact with SARS-CoV-2 positive cases (Fig 1B). The operating characteristics (ROC) curve analysis with recovered patients and healthy controls allowed to determine the cut-off values of 25, 21 and 14 IFN-γ spots forming units (sfu)/$10^6$ PBMC as positive cellular responses against the domain S1 of spike (S1), membrane (M) and nucleocapsid (N) proteins respectively, with high sensitivity and specificity (S1 Fig).

Lastly, vaccinated individuals were 28 healthcare workers, without previous SARS-CoV-2 infection (Fig 1C). They received two 30 μg doses of BNT162b2 (Pfizer-BioNTech), 21 days apart. Samples were obtained pre-vaccine, pre-boost and 15, 30 and 90 days after completing vaccination. 22/28 (79%) vaccinees were females, and the median age was 40 (interquartile range [IQR] 26–57 years).

### Development of SARS-CoV-2-specific cellular and humoral immunity during acute infection

SARS-CoV-2-specific T cells, IgG and neutralizing antibodies were sequentially analyzed in patients with natural infection, at their arrival to ER and during convalescence. Mild patients had a detectable, potent cellular immune response against S1, M and N, already within the first 2 weeks PSO (Figs 2A and S2). During this time period, a gradual increase in the specific cellular responses and positive correlations between the cellular response against the 3 proteins and days PSO (DPSO) were observed (p<0.001, Fig 2B). The peak of the SARS-CoV-2-specific

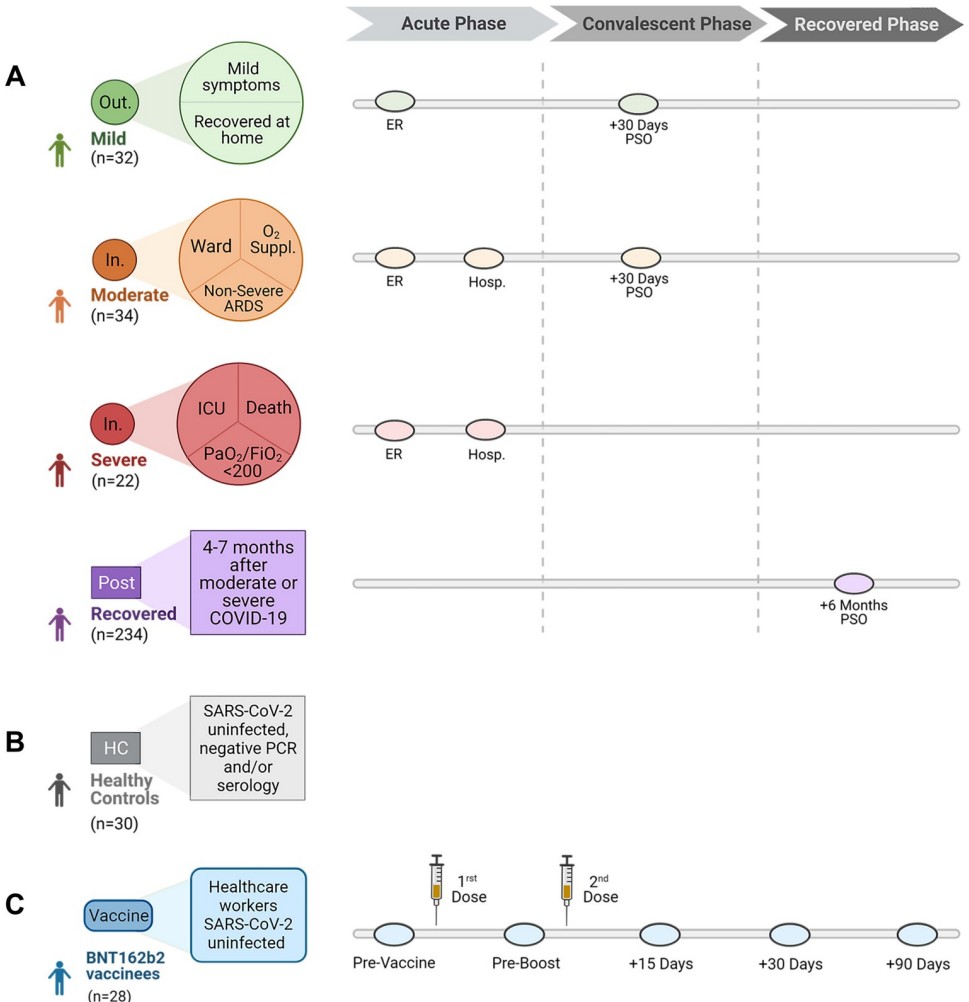

**Fig 1. Overview of patient cohorts including the time of sample collection.** (**A**) Patients with natural infection classified according to their maximum disease severity. As depicted, samples were obtained upon arrival to the emergency room (ER), a week after hospitalization (Hosp.), and during the convalescent and recovered phases depending on each cohort. (**B**) Healthy controls SARS-CoV-2 uninfected with negative PCR and/or serology. (**C**) Vaccinated individuals who received two doses of BNT162b2 21 days apart. Samples were collected pre-vaccine, pre-boost and 15, 30 and 90 days after receiving the second vaccine dose. Out.: out-patients; In.: in-patients; ICU: Intensive care unit; $PaO_2/FiO_2$: Partial pressure of arterial oxygen/Fraction of inspired oxygen ratio; PSO: Post-symptom onset. Fig 1 created with BioRender.com.

cellular response occurred 2 weeks PSO in mild patients, and although the responses to S1, M and N were generally parallel and correlated positively (Fig 2B), the peak cellular response was more potent against S1 and N than M (median of 1052, 838 and 637 peak IFN-γ sfu/$10^6$ PBMC, respectively). The cellular immune response developed upon natural SARS-CoV-2 infection gave rise to detectable IFN-γ-, interleukin (IL)-2- and bifunctional IFN-γ/IL-2-producing T cells (S3 Fig). IFN-γ and IL-2 responses were similar in magnitude and correlated repeatedly in all patient cohorts (Fig 2B). For that reason, and for the sake of brevity, we will focus on data on IFN-γ mostly. Apart from the sfu, which correspond with the frequency of antigen-specific T cell clones, we obtained the relative spot volume (RSV) which represented the amount of secreted analyte (either IFN-γ or IL-2). The total amount of secreted analyte correlated repeatedly with the frequency of specific T cells (Fig 2B). After the peak of the

**Table 1. Demographic and clinical characteristics of the cohorts.**

| | Mild (N = 32) | Moderate (N = 34) | Severe (N = 22) | Recovered (N = 234) |
|---|---|---|---|---|
| Age (years), median (IQR) | 39 (30–51) | 55 (50–67) | 64 (52–76) | 58 (48–71) |
| Female (%) | 22 (69%) | 12 (35%) | 3 (14%) | 112 (48%) |
| Maximum disease severity (WHO score) | 2 | 3–4 | 5–8 | 3–7 |
| Days from symptom onset (at emergency room or recovered sample), median (IQR) | 6 (4–10) | 8 (5–9) | 7 (4–11) | 170 (157–186) |
| Length of hospital stay (days), median (IQR) | | 5 (3–7) | 12 (8–19) | 9 (7–14) |
| COVID-19 treatment during hospitalization (%) | | | | |
| Corticosteroids | | 22 (65%) | 21 (95%) | 87 (37%) |
| Tocilizumab | | 2 (6%) | 8 (36%) | 47 (20%) |
| Remdesivir | | 2 (6%) | 11 (50%) | 4 (2%) |
| Oxygen | | 16 (47%) | 22 (100%) | 152 (65%) |
| Death (%) | | | 8 (36%) | |
| Sample collection date | Nov-Dec 2020 | Nov-Dec 2020 | Sept-Dec 2020 | August-Sept 2020 |

cellular response, SARS-CoV-2-specific T cells decreased but were still detectable during convalescence, between 21 and 45 DPSO, with similar magnitude in the responses against S1, M and N (median of 202, 300 and 178 IFN-γ sfu/$10^6$ PBMC, respectively, Fig 2A). Most mild patients did not have detectable SARS-CoV-2-specific IgG upon arrival to ER. IgG detection started on day 13 PSO and increased during convalescence ($p < 0.05$, Fig 2C). Anti-S1 IgG antibodies measured by ELISA correlated with anti-Spike IgG1 measured by a newly developed high-sensitivity flow cytometry method [31], especially during convalescence when the humoral response was fully developed ($p < 0.001$, Fig 2B). In addition to quantifying SARS-CoV-2-binding antibodies, we measured their functionality in a classical pseudovirus neutralization assay. Antibodies from mild disease patients showed a modest neutralizing capacity (Fig 2D). Both anti-S1 IgG and anti-Spike IgG1 positively correlated with the neutralizing capacity of serum samples ($r = -0.59$, $p < 0.05$ and $r = -0.78$, $p < 0.001$, correlation with normalized infection, Fig 2E).

Moderate patients had a detectable cellular immune response against S1, M and N, within the first 2 weeks PSO (Figs 3A and S2), which did not correlate with DPSO (Fig 3B). The peak of cellular immunity occurred at 21–24 DPSO, with a median of 838, 767 and 483 IFN-γ sfu/$10^6$ PBMC for S1, M and N responses respectively (Fig 3A). During convalescence (21 to 40 DPSO), S1, M and N-specific T cells remained detectable (median of 378, 370 and 107 IFN-γ sfu/$10^6$ PBMC, respectively, Fig 3A). Most moderate patients did not have detectable SARS-CoV-2-specific IgG upon arrival to ER (Fig 3C), however, they developed high antibody levels with a robust neutralizing capacity during convalescence (Fig 3D). There were strong correlations between antibody amounts and neutralization ($r = -0.79$, $p < 0.001$ for anti-S1 IgG, and $r = -0.87$, $p < 0.0001$ for anti-Spike IgG1, Fig 3E).

Severe patients had a remarkably low cellular immune response against all 3 proteins, within the first 2 weeks PSO (Figs 4A and S2), which did not correlate with DPSO either (Fig 4B). The development of the cellular response was very slow, as it had still not increased after a week of hospitalization (median of 24, 11 and 8 IFN-γ sfu/$10^6$ PBMC at ER and of 48, 25 and 33 IFN-γ sfu/$10^6$ PBMC after a week of hospitalization, for S1, M and N responses, respectively, Fig 4C). Detection of SARS-CoV-2-specific IgG upon arrival to ER was uneven: while most severe patients had undetectable IgG a few of them had already high levels (Fig 4D). During the first hospitalization week, approximately half of the patients with no antibodies developed them (Fig 4E), and those antibodies were highly neutralizing (Fig 4F). Severe patients

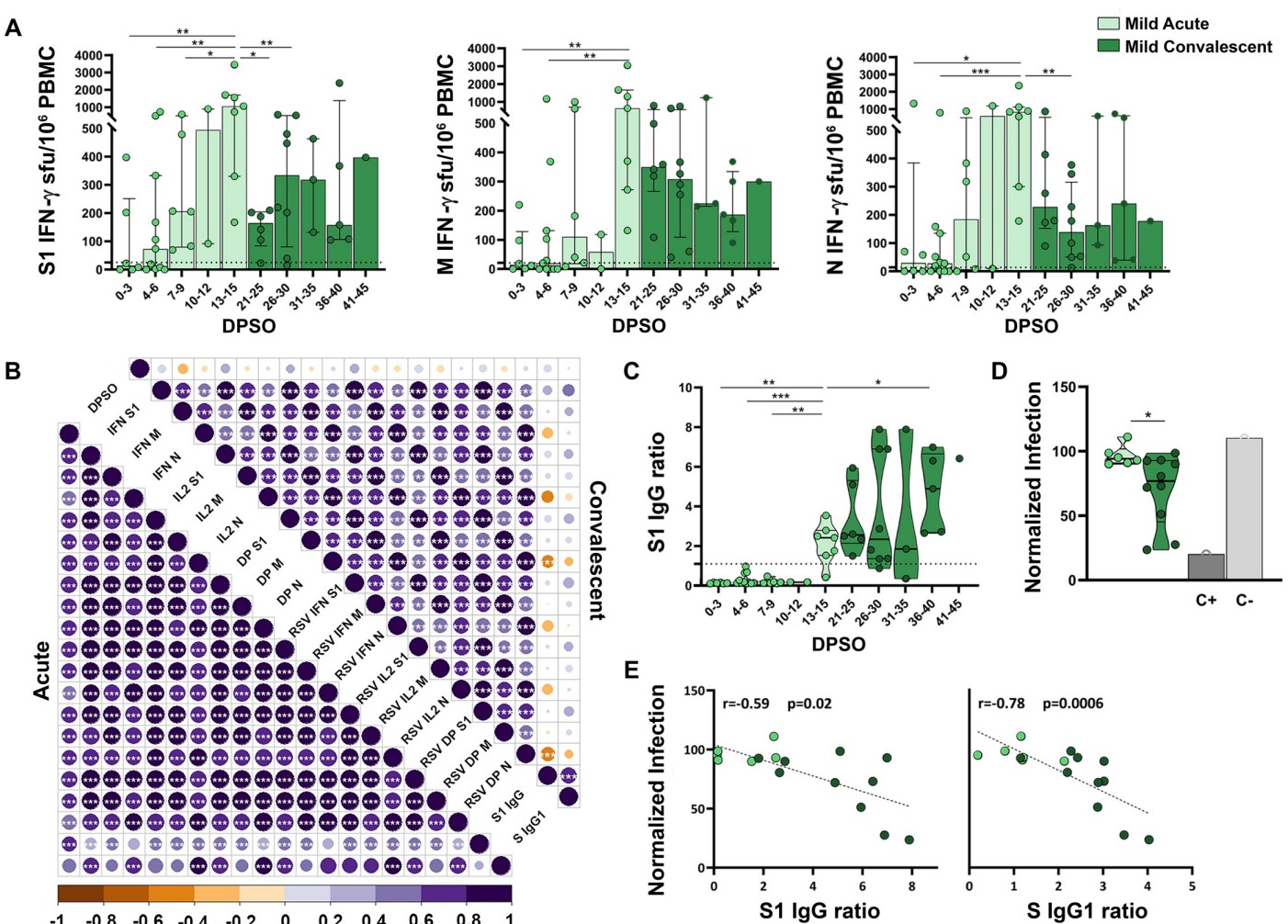

**Fig 2. Cellular and humoral immune responses in mild natural infection, during acute and convalescent phases.** (**A**) FluoroSpot IFN-γ responses against SARS-CoV-2 S1, M and N proteins according to the days post-symptom onset (DPSO). Data is represented as spot forming unit (sfu) per million PBMC. Dashed lines represent the positivity cut-off established by using a non-infected control group: >25 sfu/$10^6$ PBMC for S1, >21 sfu/$10^6$ PBMC for M and >14 sfu/$10^6$ PBMC for N. (**B**) Correlation matrices of cellular and humoral immune variables analyzed. Only significant correlations (p<0.05) are represented with asterisks. Positive correlations appear in purple, and negative correlations appear in orange. The size and the colour gradient of the circle corresponds to the magnitude of the correlation. (**C**) Results of SARS-CoV-2 S1 IgG ELISA according to DPSO. Dashed line represents the established cut-off of positivity (ratio ≥1.1). (**D**) Neutralizing capacity of 1/500 diluted sera, tested with S protein-pseudotyped virus represented as percentage of normalized infection neutralized. (**E**) Correlation between neutralizing capacity and the semi-quantitative results of SARS-CoV-2 IgG ELISA (left) and anti-Spike IgG1 assessed by flow cytometry (right). Linear regressions were performed using Spearman's rank test. Horizontal bars and whiskers represent median values and interquartile ranges, respectively. The significance between groups was determined using Mann Whitney or Wilcoxon signed rank tests, *p<0.05, **p<0.01, ***p<0.001, ****p<0.0001. RSV: Relative Spot Volume; DP: IFN-γ/IL-2 double positive sfu.

showed the strongest correlation between levels of antibodies and neutralizing response (r = -0.90, p<0.001 for anti-S1 IgG, and r = -0.87, p<0.001 for anti-Spike IgG1, Fig 4G). Eight of the 22 (36%) severe patients died during hospitalization. These patients who died had significantly lower cellular response against all 3 proteins (Fig 4H) and a trend towards lower humoral response (Fig 4I) than severe patients who survived. In summary, upon arrival to ER, severe patients showed a negative correlation between antibody levels and some compartments of the cellular response, which reflected a discoordination between both arms of the immune response and the dominance of the humoral over the cellular immunity (Fig 4B).

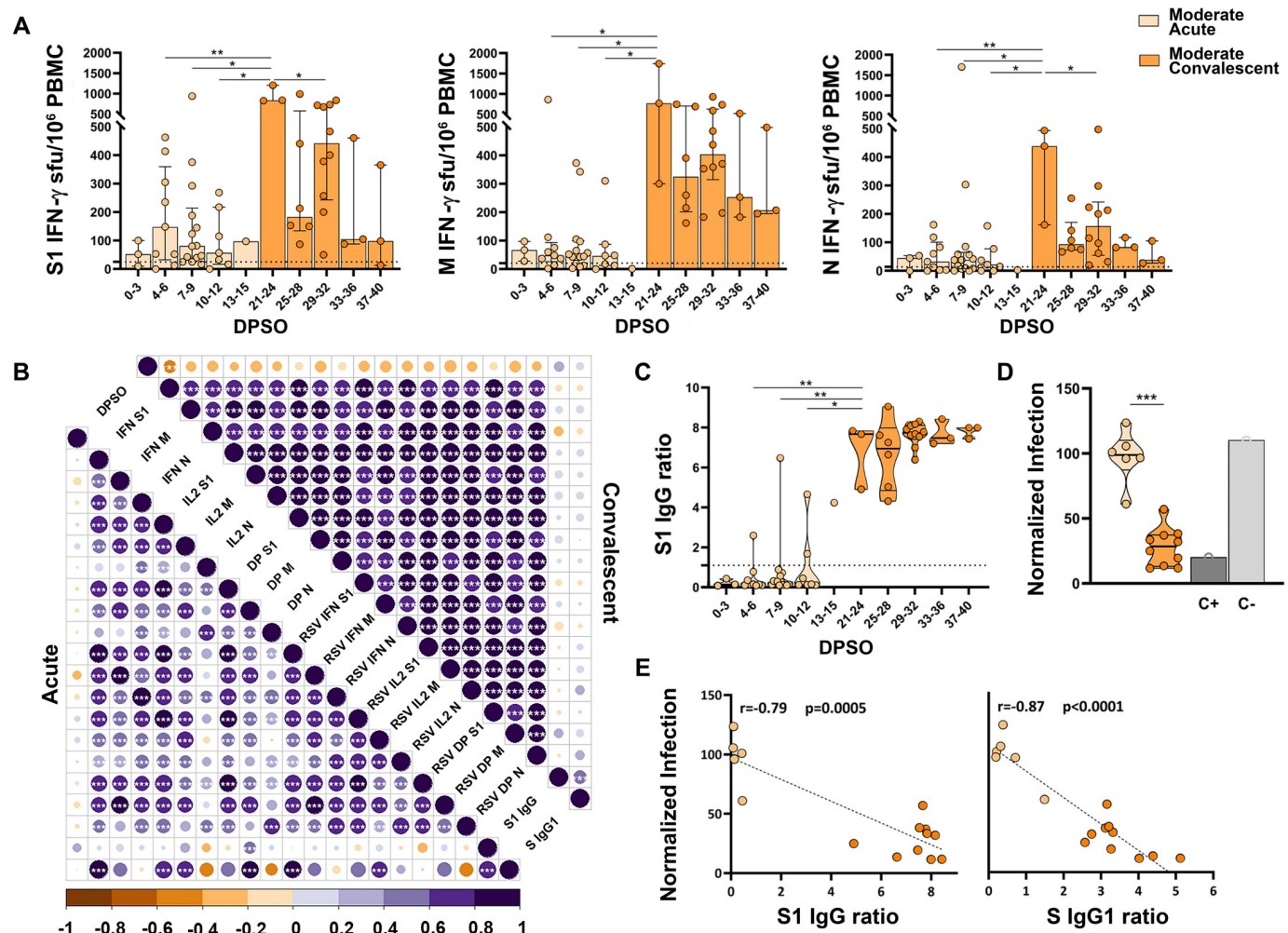

**Fig 3. Cellular and humoral immune responses in moderate natural infection, during acute and convalescent phases.** (**A**) FluoroSpot IFN-γ responses against SARS-CoV-2 S1, M and N proteins according to the days post-symptom onset (DPSO). Dashed lines represent the positivity cut-off established by using a non-infected control group: >25 sfu/$10^6$ PBMC for S1, >21 sfu/$10^6$ PBMC for M and >14 sfu/$10^6$ PBMC for N. (**B**) Correlation matrices of cellular and humoral immune response variables analyzed. Only significant correlations (p<0.05) are represented with asterisks. (**C**) Results of SARS-CoV-2 S1 IgG ELISA according to DPSO. (**D**) Neutralizing capacity tested with S protein-pseudotyped virus represented as percentage of normalized infection neutralized. (**E**) Correlation between neutralizing capacity and the semi-quantitative results of SARS-CoV-2 IgG ELISA (left) and anti-Spike IgG1 assessed by flow cytometry (right). Linear regressions were performed using Spearman's rank test. Horizontal bars and whiskers represent median values and interquartile ranges, respectively. The significance between groups was determined using Mann Whitney or Wilcoxon signed rank tests, *p<0.05, **p<0.01, ***p<0.001, ****p<0.0001. (See Fig 2 footnote for more detailed information).

## Association between dynamics of SARS-CoV-2-specific immune responses and COVID-19 severity

We next compared the dynamics of SARS-CoV-2 immunity among the 3 cohorts with different severity. Specific T cells started to increase in mild and moderate patients in the first week of symptoms, when they were barely detectable in severe patients. During the second week PSO, the cellular response developed robustly in mild patients with a significant increase between the two weeks (p<0.01, Fig 5A), while it remained unchanged in moderate and severe patients. From the start of infection, we observed that the cellular response developed faster in mild patients, reaching their peak approximately 14 DPSO (Fig 2A), than in moderate patients, who reached their peak response a week later (Fig 3A). Nonetheless, the magnitude of the peak

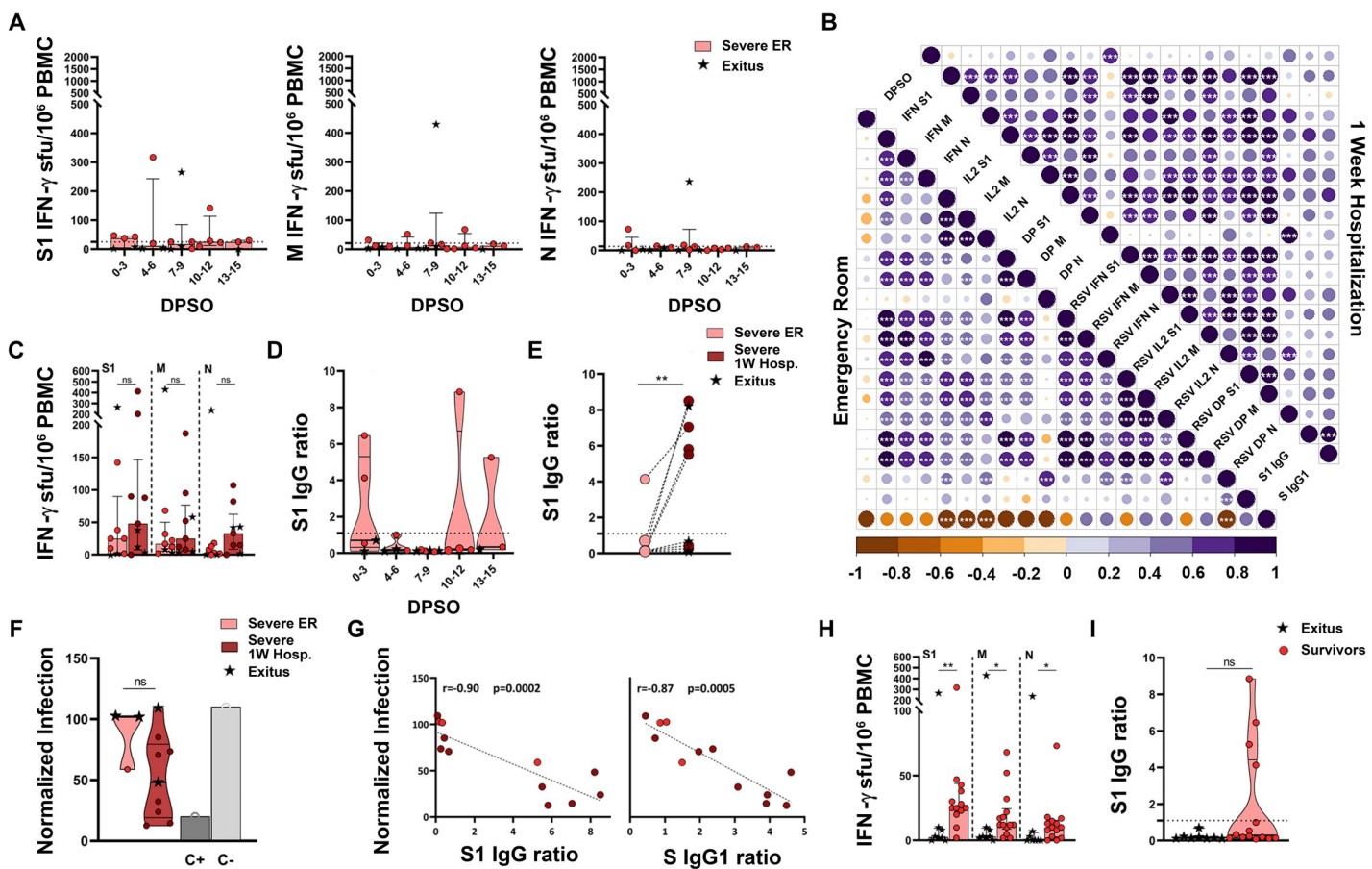

**Fig 4. Cellular and humoral immune responses in severe natural infection, during acute phase, measured at emergency room (ER) and a week after hospitalization (1W Hosp.).** (**A**) FluoroSpot IFN-γ responses against SARS-CoV-2 S1, M and N proteins according to the days post-symptom onset (DPSO). Dashed lines represent the positivity cut-off established by using a non-infected control group: >25 sfu/10⁶ PBMC for S1, >21 sfu/10⁶ PBMC for M and >14 sfu/10⁶ PBMC for N. (**B**) Correlation matrices of cellular and humoral immune variables analyzed. Only significant correlations (p<0.05) are represented with asterisks. (**C**) FluoroSpot IFN-γ responses against SARS-CoV-2 S1, M and N proteins at ER and a week after hospitalization. (**D-E**) Results of SARS-CoV-2 S1 IgG ELISA according to DPSO. (**F**) Neutralizing capacity tested with S protein-pseudotyped virus represented as percentage of normalized infection neutralized. (**G**) Correlation between neutralizing capacity and the semi-quantitative results of SARS-CoV-2 IgG ELISA (left) and anti-Spike IgG1 assessed by flow cytometry (right). (**H-I**) Comparison of S1, M and N T-cell (H) and anti-S1 IgG (I) responses between survivor and non-survivor patients (black stars). Linear regressions were performed using Spearman's rank test. Horizontal bars and whiskers represent median values and interquartile ranges, respectively. The significance between groups was determined using Mann Whitney, Wilcoxon signed rank or Kruskal-Wallis tests, *p<0.05, **p<0.01, ***p<0.001, ****p<0.0001. (See Fig 2 footnote for more detailed information).

S1, M and N-specific cellular response was similar in mild and moderate patients. Severe patients, however, had not started developing high numbers of specific T cells by the end of the acute infection follow-up, by 21 DPSO (Fig 4C). In addition to developing specific T cell clones faster and in higher numbers, specific T cells from milder patients were more functional than those from severe patients, as shown by a higher amount of IFN-γ and IL-2 secreted per clone (Figs 5B and S3). In our cohort, we did not observe an association between viral load at diagnosis and disease severity (Fig 5C). Nevertheless, we found a strong correlation, mostly in moderate patients (r = 0,98, p<0.0001), between a more potent specific T cell response and a higher Ct value at diagnosis, suggesting that patients with stronger specific T cell responses are more able to control viral replication (Fig 5D). These results confirm and further validate smaller studies which had indicated that the development of a functional T cell response is associated with mild disease.

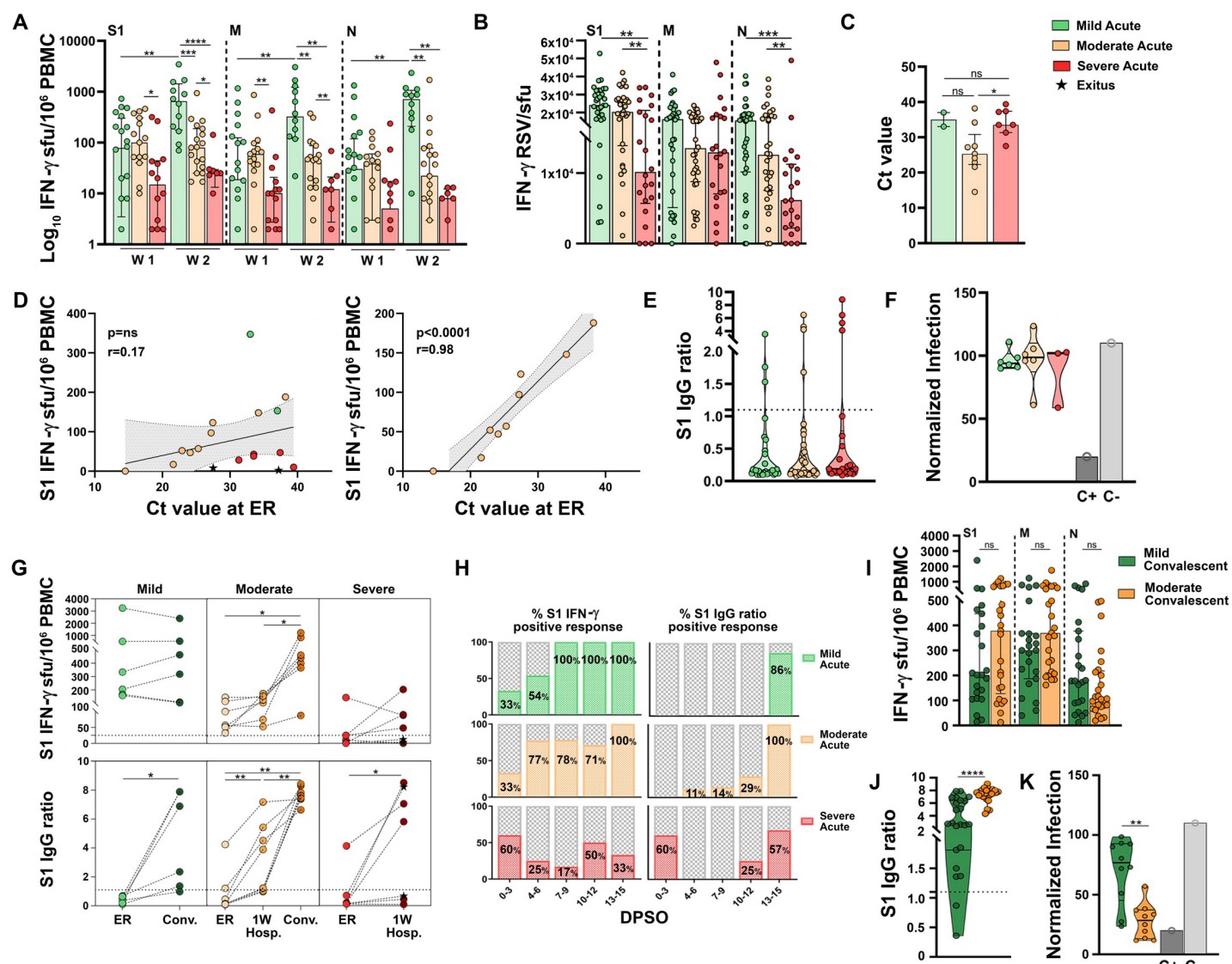

**Fig 5. Association between dynamics of SARS-CoV-2-specific immune responses and COVID-19 severity.** (**A**) SARS-CoV-2-specific IFN-γ-producing T-cell responses reactive to the S1, M and N proteins in mild, moderate, and severe patients during the first (W1) and the second (W2) week post-symptom onset. (**B**) Relative Spot Volume (RSV) per spot of secreted IFN-γ after S1, M and N peptide pool stimulation in mild, moderate, and severe patients during acute infection. (**C**) Ct values obtained on real time RT-PCR for detection of the E gene upon arrival to ER in mild, moderate and severe patients. A higher Ct value corresponded to a lower viral load. (**D**) Correlation between S1 IFN-γ-producing T-cells and the relative viral load represented as Ct value at ER. (**E**) Semi-quantitative results of SARS-CoV-2 S1 IgG ELISA in the acute phase in mild, moderate, and severe patients. Dashed line represents the established cut-off of positivity (ratio ≥1.1). (**F**) Neutralizing capacity tested with S protein-pseudotyped virus represented as normalized infection. (**G**) Longitudinal data on the dynamics of cellular and humoral responses in six representative patients from the 3 severity cohorts. (**H**) Percentage of positive S1 cellular and humoral response during acute phase according to the days post symptom onset (DPSO) in mild, moderate and severe patients. (**I**) Comparison of IFN-γ responses against SARS-CoV-2 S1, M and N proteins between mild and moderate patients in convalescent phase. (**J**) Comparison of semi-quantitative results of SARS-CoV-2 S1 IgG between mild and moderate patients in convalescent phase. (**K**) Comparison of neutralizing capacity tested with S protein-pseudotyped virus represented as normalized infection. Horizontal bars and whiskers represent median values and interquartile ranges, respectively. Linear regressions were performed using Spearman's rank test. The significance between groups was determined using Mann Whitney, Wilcoxon signed rank or Kruskal-Wallis tests, *p<0.05, **p<0.01, ***p<0.001, ****p<0.0001. ER: Emergency Room; Conv: Convalescence; 1W Hosp.: 1 Week Hospitalization; DPSO: Days post-symptom onset.

Both SARS-CoV-2-specific IgG and neutralization were mostly undetectable in all 3 cohorts during the first 2 weeks of symptoms (Fig 5E and 5F). The different dynamics of cellular and humoral immune response development are shown with 6 representative patients from each of the 3 cohorts in Fig 5G. On the whole, the adaptive immune response seemed to be more

coordinated in mild and moderate patients, as most patients had developed a positive cellular response within the first 2 weeks PSO (23/32 [72%] and 26/34 [76%] respectively, Fig 5H), which was followed by the development of virus-specific IgG (86% mild and 100% moderate patients had detectable antibodies by day 13–15 PSO). Most severe patients, however, had not developed a positive SARS-CoV-2-specific cellular response within the first 2 weeks PSO, they lacked any correlation between positivity of the cellular response and DPSO when they attended the ER, and the development of virus-specific IgG did not follow a temporal pattern during acute infection (Fig 5H). By investigating if the SARS-CoV-2 specific cellular response could have a prognostic value, we found that a higher frequency of ER-measured specific T cells reduced the probability of developing severe COVID-19 (odds ratio [OR] per 100 IFN-γ sfu/$10^6$ PBMC increment: 0.45, 95%CI 0.20–0.76, p<0.05). Given that age and sex differed among mild, moderate and severe patients (Table 1), and that both older age and male sex have been associated with disease severity [1–3], we performed a multivariate analysis to test if the decreased risk of developing severe COVID-19 associated to high frequency of specific T cells could be influenced by these two variables. Independently of age and sex, the S1-specific cellular response measured at ER remained as a protective factor (OR per 100 IFN-γ sfu/$10^6$ PBMC increment: 0.47, 95%CI 0.20–0.87, p<0.05), while age and male sex appeared as independent COVID-19 severity risk factors (OR: 1.05, 95%CI 1.01–1.10, p = 0.01 and OR: 5.02, 95%CI 1.31–26.07, p<0.05, respectively).

During convalescence, there was no significant difference in the cellular response against all 3 proteins between mild and moderate patients (Fig 5I). On the contrary, antibodies were much higher in moderate than in mild patients (p<0.0001, Fig 5J). Accordingly, mild patients showed modest neutralization capacity while moderate patients had a significantly more robust neutralizing response (p<0.01, Fig 5K).

SARS-CoV-2 infection is known to cause lymphopenia, which associates with COVID-19 severity [32,33]. We found lymphopenia in acute moderate patients, mainly due to reduced T cells (S1 Table). There was a substantial decrease in CD4+ and CD8+ T cells, and a milder reduction in B cells. All lymphocyte populations recovered during convalescence. Severe patients had a profound lymphopenia during acute infection, with a reduction in CD4+, CD8+, B and NK cells. In addition, the overall T cell functionality was clearly reduced in severe patients when cells were polyclonally stimulated with anti-CD3/anti-CD28 (S4 Fig). We asked if the lymphopenia and/or the suboptimal global T cell response during acute infection could be responsible for the lack of SARS-CoV-2-specific T cells in severe patients. When we adjusted in each sample the specific cellular response by the number of sfu/$10^6$ PBMC after polyclonal stimulus, the results showed again how mild patients developed a significantly faster and more potent SARS-CoV-2-specific cellular response than hospitalized patients in the beginning of the acute infection (S4 Fig), suggesting that the profound lymphopenia and general reduction in T cell response in severe patients could only partially contribute to the lack of SARS-CoV-2-specific cellular response.

## Duration of SARS-CoV-2-specific cellular and humoral immunity after natural infection

In order to assess the duration of adaptive immunity after natural infection, we analyzed the cohort of 234 COVID-19 recovered patients (Fig 1A). Most recovered patients maintained detectable virus-specific memory T cells, significantly more against S1 than M and N (median of 140, 117 and 75 IFN-γ sfu/$10^6$ PBMC respectively, p<0.05, Fig 6A). In total, 227/234 (97%) patients maintained a cellular response against S1, M and/or N above the positivity threshold. Unlike in the acute and convalescent phase, the number of IFN-γ- and IL-2-producing T cells differed in recovered patients. Six months PSO we detected significantly more IL-2-producing

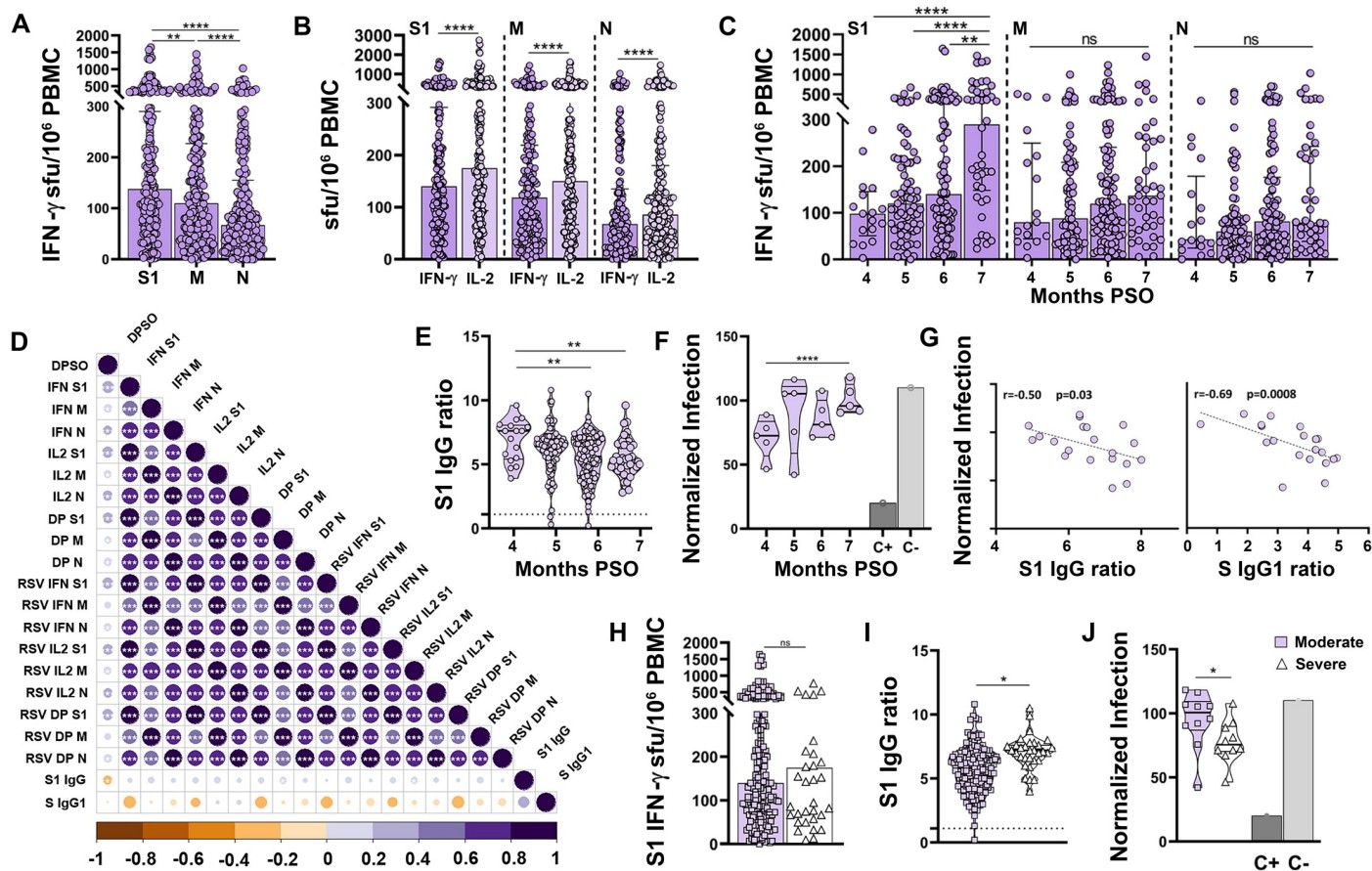

**Fig 6. Duration of SARS-CoV-2-specific cellular and humoral immunity after natural infection.** (**A**) S1, M and N cellular responses in recovered patients. (**B**) Comparison between IFN-γ and IL-2-producing T-cell responses. (**C**) IFN-γ-producing T-cell responses according to the months post-symptom onset (PSO). (**D**) Correlation matrix of cellular and humoral immune variables analyzed. Only significant correlations (p<0.05) are represented with asterisks. (**E**) SARS-CoV-2 S1 IgG according to the months PSO. Dashed line represents the established cut-off of positivity (ratio ≥1.1). (**F**) Neutralizing capacity represented as normalized infection according to the months PSO. (**G**) Correlation between neutralizing capacity and the semi-quantitative results of SARS-CoV-2 IgG ELISA (left) and anti-Spike IgG1 measured by flow cytometry. (**H-J**) S1 IFN-γ-producing T-cells (H), semi-quantitative results of SARS-CoV-2 S1 IgG (I) and neutralizing capacity (J) in recovered patients who suffered a moderate or severe acute disease. Linear regressions were performed using Spearman's rank test. Horizontal bars and whiskers represent median values and interquartile ranges, respectively. The significance between groups was determined using Mann Whitney or Kruskal-Wallis tests, *p<0.05, **p<0.01, ***p<0.001, ****p<0.0001. RSV: Relative Spot Volume; DP: IFN-γ/IL-2 double positive sfu.

than IFN-γ-producing clones (174 vs 140, 152 vs 117 and 92 vs 75 sfu/10[6] PBMC against S1, M and N, respectively, all p<0.0001, Fig 6B). Moreover, we found an unexpected gradual increase in the cellular response from month 4 to month 7 PSO, which was significant for S1 (p<0.01 Fig 6C), and positive correlations between the cellular response and DPSO (Fig 6D). Recovered patients had normalized their lymphocyte populations (S1 Table), however they showed an increased polyclonal T cell response compared to healthy controls (p<0.001, S5 Fig), which augmented with time after infection. After adjusting by the sfu/10[6] PBMC values of polyclonal responses, the increase in SARS-CoV-2-specific cellular response was considerably reduced, suggesting that the observed increase in SARS-CoV-2-specific cellular response from month 4 to 7 PSO was likely due to a general increment in T cell functionality rather than to a specific boost of cellular response (S5 Fig).

A robust humoral response anti-COVID-19 was still detected in recovered patients, as 99% (231/234) of them were positive for SARS-CoV-2-specific IgG. Contrarily to the cellular response, there was a gradual significant decline from month 4 to month 7 in antibodies

(p<0.01, Fig 6E) and neutralizing capacity (p<0.0001, Fig 6F), and a negative correlation was observed between anti-S1 IgG levels and DPSO (Fig 6D). Antibody levels paralleled the neutralizing capacity (r = -0.50, p<0.05 for anti-S1 IgG and r = -0.69, p<0.001 for anti-Spike IgG1, Fig 6G), although these correlations were weaker than in acute and convalescent patients (Figs 2E, 3E and 4G).

There was no difference in memory cellular response between recovered patients who had suffered a moderate (N = 203) or a severe (N = 31) COVID-19 form (Fig 6H), while subjects recovered from severe COVID-19 maintained significantly higher antibody levels (p<0.05, Fig 6I), with higher neutralization capacity (p<0.05, Fig 6J). The absence of specific T cells in patients with acute severe infection (Fig 4), which were detectable in subjects recovered from severe COVID-19 suggests that severe patients may develop a specific cellular response sometime during convalescence.

## Development of SARS-CoV-2-specific cellular and humoral immunity after BNT162b2 vaccination

We studied the development of the adaptive immunity after vaccination in a cohort of individuals naïve for SARS-CoV-2 infection. After the first vaccine dose, 82% (23/28) of vaccinees developed a positive cellular response, which as expected was specific against S1 (median of 140 IFN-$\gamma$ sfu/$10^6$ PBMC, Fig 7A). The administration of the second boost vaccine dose potentiated the cellular response, which peaked 2 weeks after full vaccination (median of 621 IFN-$\gamma$ sfu/$10^6$ PBMC) and then decreased steadily a month and three months post-vaccination (median of 439 and 158 IFN-$\gamma$ sfu/$10^6$ PBMC, respectively, Fig 7B). Despite this decrease, all vaccinees except one remained above the positivity threshold. A similar dynamic was observed in the development of IL-2-producing SARS-CoV-2-specific T cells and IFN-$\gamma$/IL-2 double positive T cells (Fig 7C and 7D). There was a strong positive correlation between IFN-$\gamma$, IL-2 and bifunctional T cell response, measured as the number of specific-T cell clones (sfu) and as the total amount of cytokine secreted (RSV) after the first and second vaccine doses (Fig 7E). Additionally, as reported in the phase I/II clinical trial for the vaccine candidate BNT162b1 [22], we confirmed that the BNT162b2 vaccine elicited exclusively Th1 T cell responses as noted by the increased frequency of CD4 Th1 PD1+, but not CD4 Th2 or CD4 Th17 PD1+, cells 2 weeks after full vaccination (p<0.01, Fig 7F).

The development of anti-S1 antibodies after vaccination paralleled the dynamics of the cellular response. A single vaccine dose generated detectable IgG in 94% (26/28) of vaccinees (Fig 7G). Two weeks after completing vaccination, there were extremely high antibody levels in all individuals, which remained stable a month after vaccination. Three months after vaccination there was a stark decrease in IgG levels (p<0.0001, Fig 7G), although all vaccinees remained well above the positivity threshold. As described by the vaccine trials [21,22], a single dose of vaccine was insufficient to develop neutralizing antibodies (Fig 7H). After the boost dose, sera from vaccinees were able to partially neutralize infection, although the neutralization capacity decreased already at a month, compared to two weeks, after full vaccination (p<0.01, Fig 7H). Similar to natural infection, there was a correlation between levels of antibodies and neutralizing response (r = -0.88, p<0.0001 for anti-S1 IgG, and r = -0.70, p<0.0001 for anti-Spike IgG1, Fig 7I). Given the parallel cellular and humoral response to the vaccine, a month after full vaccination, there was a positive correlation between the two arms of the adaptive immunity (Fig 7E). While no relationship between age and magnitude of the cellular response was observed (Fig 7J), age negatively correlated with antibody levels (r = -0.56, p<0.01, Fig 7K), despite of which all vaccinees showed high levels of anti-S1 IgG, with no differences in neutralizing capacity among subjects of different ages (Fig 7L).

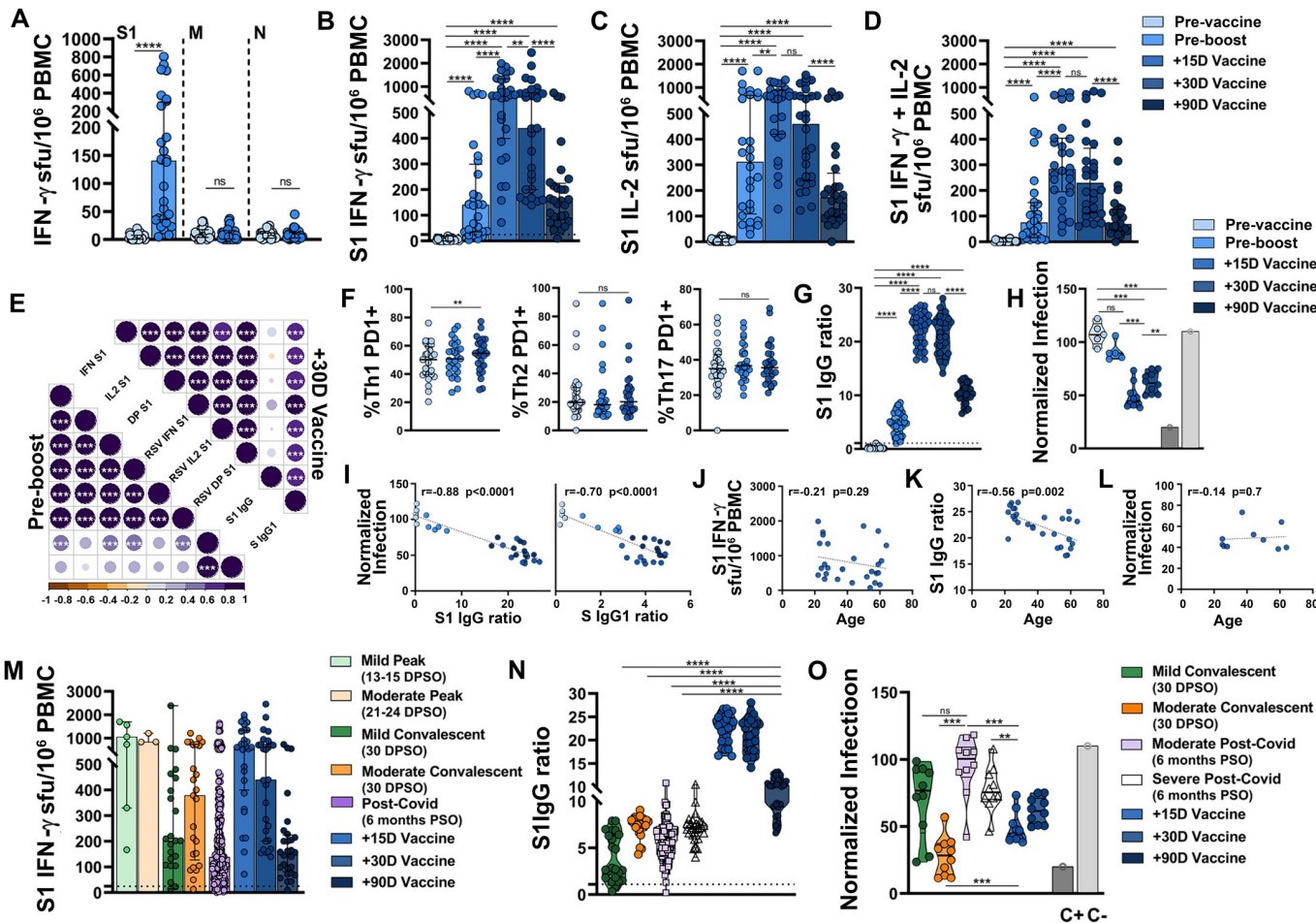

**Fig 7. Development of SARS-CoV-2-specific cellular and humoral immunity after BNT162b2 vaccination.** (**A**) IFN-γ-producing T-cell responses against S1, M and N proteins pre-vaccine and pre-boost. (**B-D**) S1 SARS-CoV-2-specific IFN-γ (B), IL-2 (C) and both cytokines (D) producing T-cells pre-vaccine, pre-boost and 15, 30 and 90 days after complete vaccination. Dashed line represents the positivity cut-off: >25 IFN-γ sfu/10⁶ PBMC. (**E**) Correlation matrix of adaptive immune variables analyzed. Only significant correlations (p<0.05) are represented with asterisks. (**F**) Frequency of CD4 Th1, Th2 and Th17 PD1+ cells pre-vaccine, pre-boost and 15 days after complete vaccination. (**G**) Anti-S1 IgG results pre-vaccine, pre-boost and 15, 30 and 90 days after complete vaccination. Dashed line represents the positivity cut-off (ratio ≥1.1). (**H**) Neutralization capacity pre-vaccine, pre-boost and 15 and 30 days after complete vaccination. (**I**) Correlation between neutralization and anti-S1 IgG (left) and anti-Spike IgG1 (right). (**J-L**) Correlation between age and S1 IFN-γ-producing T-cells (J), anti-S1 IgG (K) and neutralization (L) 15 days after full vaccination when the vaccine-elicited immune response peaked. (**M**) Comparison of S1 T-cell response between SARS-CoV-2 natural infection and complete vaccination. (**N-O**) Comparison of anti-S1 IgG (N) and neutralization (O) between SARS-CoV-2 natural infection and complete vaccination. Linear regressions were performed using Spearman's rank test. Horizontal bars and whiskers represent median values and interquartile ranges, respectively. The significance between groups was determined using Mann Whitney, Wilcoxon signed rank or Kruskal-Wallis tests, *p<0.05, **p<0.01, ***p<0.001, ****p<0.0001. DPSO: Days post symptom onset.

## Comparison of SARS-CoV-2-specific immunity development after natural infection or real-world vaccination

Finally, we compared the magnitude and dynamics of the adaptive immune response after vaccination to that developed after natural infection. The peak of the cellular response 2 weeks after complete vaccination was comparable to the peak response in acute mild and moderate patients, 2 to 3 weeks PSO (Fig 7M). After the peak, the cellular response started to decline and 3 months after vaccination it was similar to that of COVID-19 recovered patients 4–7 months after infection. The production of anti-S1 IgG reached much higher levels in vaccinees than in naturally infected individuals, either convalescent or recovered patients, up to three months

post-vaccination (p<0.0001, Fig 7N). However, the neutralization capacity of the antibodies elicited by the vaccine was relatively lower than that of antibodies elicited by the natural infection. Despite having higher levels of IgG, the neutralization capacity of vaccinees was intermediate between that of mild and moderate convalescent patients (Fig 7O).

## Discussion

Our analysis of cellular and humoral SARS-CoV-2-specific immunity in patients with varying degrees of COVID-19 severity indicates that SARS-CoV-2-specific T cells are essential for disease control and that the delayed mounting of SARS-CoV-2-specific cellular immune response is associated with severe infection course. In addition, we describe in real-world conditions the development of a robust, coordinated cellular and humoral immune response with the BNT162b2 mRNA COVID-19 vaccine, up to three months after completing the 2-dose vaccination schedule.

To date, this is the largest study analyzing the SARS-CoV-2-specific cellular and humoral immune response in patients upon ER arrival, which allowed the characterization of both arms of adaptive immunity from the start of natural infection without any interference from immunomodulatory medication. The timing in which the different responsive compartments of immunity appeared was crucial for determining the course of the infection. Patients who developed a prompt IFN-γ- and IL-2-producing cellular response followed by antibody production had a mild COVID-19 course, an observation in agreement with the well-known concept that the early development of virus-specific T cells and the subsequent production of neutralizing antibodies are required for effective control of viral infections [34]. On the other extreme, patients who did not develop an early specific cellular response and initiated a humoral immune response with subsequent production of high levels of antibodies, were not able to eliminate the virus efficiently and developed severe symptoms. A strong humoral response in severe COVID-19 patients has already been reported [11,16,35]. We observed that the inadequate adaptive immune response developed in severe patients could be explained partially by the profound lymphopenia and general decline in the T cell response. However, it is likely that the differences observed in the adaptive immune response between mild, moderate and severe COVID-19 forms also derive from differences in the patients' initial innate immune response. In particular, type I interferons, which are important for anti-viral host defence and may be required to initiate cell-mediated immune responses [36], could be hindered by genetic defects or autoantibodies that have been associated with severe COVID-19 [37,38]. Antibodies, and their neutralizing capacity, were not associated with reduced disease severity in our acute patients, which was also observed in Middle East respiratory syndrome coronavirus (MERS-CoV)-infected patients [39]. Antibodies which may not be efficient at clearing the acute viral infection, may well provide protection against infection when present before exposure to the virus [40,41].

In our study, robust antigen-specific cellular and IgG responses developed concomitantly after vaccination, unlike in natural infection. The fast and potent immunity triggered by this mRNA vaccine explains its high effectiveness in preventing SARS-CoV-2 infection in the real world [23,24]. As observed in the clinical trial by Sahin *et al.* [22], the BNT162b2 vaccine elicited higher antibody levels, although with inferior neutralizing capacity than those formed after suffering moderate COVID-19. A current concern is the immunogenicity of SARS-CoV-2 vaccines in elderly. Our results in adults 22 to 64 years old show no negative impact of age in the cellular response elicited by the vaccine and a decrease in the amount of SARS-CoV-2-specific IgG, but not in neutralizing activity, associated with age. While some studies report weak humoral and cellular immunogenicity of the BNT162b2 vaccine in COVID-19 uninfected

nursing home residents [42], others show optimal antibody titers irrespective of age [43] and, more importantly, a reduction in SARS-CoV-2 infections in vaccinated elderly [44]. Finally, three months after complete vaccination, the SARS-CoV-2-specific cellular response was similar, and the antibody levels were higher than in COVID-19 recovered patients. This coincides with the recent report by Tan et al [45] in which a similar magnitude of T cell responses between the vaccinees and recovered patients was observed three months after vaccination or infection. It could therefore be speculated that the protection due to immunization would last similarly to that conferred by natural infection. Studies on the long-term immunogenicity and effectiveness of COVID-19 vaccines will be informative to design vaccination strategies in the future. So far, several publications have investigated the durability of vaccine-induced humoral responses up to 6 months after vaccination. A gradual and substantial decrease in IgG specific antibodies and neutralizing antibody levels has been reported through the first half a year post-vaccination [28,46,47], independently of having received the mRNA based BNT162b2 or the adenoviral vector-based ChAdOx1 nCoV-19 [48]. Both, BNT162b2-induced SARS-CoV-2 antibody and T-cell responses have been shown to experience a more important decline after 6 months in the elderly than in young health-care workers [49]. Although these observations support the recommendation for booster vaccinations, the effect of antibody waning needs to be clearly determined, as, importantly, specific memory B cells remain 6 months after the immunization [29] or after the natural infection, even in the absence of detectable neutralizing antibodies [50].

A robust SARS-CoV-2-specific cellular and humoral immunity remained months after recovery. We initially observed an increase in SARS-CoV-2-specific T cells from month 4 to month 7 PSO, which paralleled the general increase in T cell functionality upon immune reconstitution following severe lymphopenia. The relativization of the specific response to the overall T cell function allowed a more accurate interpretation of the results obtained. Importantly, SARS-CoV-2-specific T cell immunity remained stable from month 4 to month 7, while specific IgG declined over this period. This information from recovered patients further supports our finding in acute infection that specific T cells are essential in infection control and will likely be essential for long-term COVID-19 protection. Furthermore, Sekine *et al.* [51] described the presence of SARS-CoV-2-specific T cells in seronegative exposed and convalescent individuals, implying that only measuring specific IgG may underestimate protection against COVID-19. In addition, recovered patients had significantly more IL-2-producing SARS-CoV-2-specific T cells than IFN-γ-producing ones. The higher prevalence of IL-2-producing SARS-CoV-2-specific cell clones could support the predominance of CD4+, rather than CD8+, T lymphocytes as expression of immune memory in patients who overcome COVID-19 [35].

A limitation of this study is that the FluoroSpot method does not differentiate between CD4- and CD8-SARS-CoV-2-specific T cells. Further studies to understand the degree of each cell subset participation in the response after infection or vaccination will be important. It should also be noted that the T cell response was analyzed using peptides covering the S1 region, which contains the receptor binding domain, but the potential response against S2 region was not analyzed. In addition, we proposed that severe disease develops because of the lack of specific cellular response to contain the virus. Elucidating the mechanisms leading to the failure in mounting an early, robust immune response against SARS-CoV-2 in some patients may help to develop new therapies and improve their prognosis.

In summary, our results not only confirm that the prompt induction of SARS-CoV-2-specific T cells in naturally infected patients is essential for disease control, but also that the SARS-CoV-2 specific cellular response measured at ER has a prognostic value as it is an age- and sex-independent protective factor against developing severe COVID-19. In addition,

specific cellular responses remained stable in recovered patients and were present up to three months after vaccination. These results highlight the relevance of monitoring SARS-CoV-2-specific cellular immune responses, and not only antibody levels, for prognosis in natural infection, and as a correlate for protection after infection and after vaccination.

## Materials and methods

### Ethics statement

All participants were enrolled after signing the informed written consent of two studies approved by the institutional clinical research ethics committee (references 21/039 and 21/056).

### Study design

A total of 380 subjects were recruited at Hospital Universitario 12 de Octubre (Madrid, Spain). These subjects belonged to six different cohorts (see Fig 1A and Study design and participants section in Results). Natural infection was confirmed by a positive SARS-CoV-2 real time reverse transcription polymerase chain reaction (RT-PCR) or a SARS-CoV-2 antigen test in all COVID-19 patients.

### FluoroSpot assay

Peripheral blood mononuclear cells (PBMCs) were freshly isolated by density-gradient centrifugation using Ficoll-Paque within a maximum of 12 hours from blood being drawn and were seeded in duplicate at 300,000 cells/well in IFN-γ IL-2 FluoroSpot plates (MabTech). Cells were supplemented with 15-mer overlapping peptides covering the S1 domain of the S glycoprotein (SARS-CoV-2 S1 scanning pool, MabTech), the N protein (Epitope Mapping Peptide Set [EMPS] SARS-CoV-2 NCAP-1, JPT), and the M protein (EMPS SARS-CoV-2 VME1, JPT) at a final concentration of 1 μg/mL. Negative control wells contained anti-CD28 mAb (1 μg/mL) and positive control wells included anti-CD3 mAb (MabTech). Assays were incubated for 16–18 hours at 37°C. Spots were counted using an automated IRIS FluoroSpot Reader System (MabTech). To quantify antigen-specific responses, spots of the negative control wells were subtracted from the mean spots test wells, and the results were expressed as IFN-γ or IL-2 producing spot forming units (sfu) per $10^6$ PBMCs. Results were excluded if negative control wells had >80 IFN-γ sfu/$10^6$ PBMCs or positive control wells had <400 IFN-γ sfu/$10^6$ PBMCs. Reponses were considered positive if the results were at least three times higher than the mean of the negative control wells and above of the following antigen-specific cut-off values: >25 IFN-γ sfu/$10^6$ PBMCs for the S1 domain of the S glycoprotein, >14 IFN-γ sfu/$10^6$ PBMCs for the N protein, and >21 IFN-γ sfu/$10^6$ PBMCs for the M protein, as previously published [52].

### Anti-S1 IgG detection by ELISA

Serum SARS-CoV-2 IgG antibodies targeting the S1 protein were detected with the Euroimmun Anti-SARS-CoV-2 enzyme-linked immunosorbent assay (ELISA) (Euroimmun AG, Lübeck, Germany) according to manufacturer's instructions. Optical density (OD) values were measured at 450 nm using the PR 3100 microplate reader (Bio-Rad Life Science, Marnes-La-Coquette, France). Results were evaluated semi-quantitatively by calculating the ratio of the OD value of the sample over the OD value of the calibrator (relative OD), with the following cut-off values: ratio <0.8: negative; ratio ≥ 0.8 to <1.1: borderline; and ratio ≥1.1: positive.

## Anti-Spike IgG1 detection by flow cytometry

Jurkat cells co-expressing full-lenght Spike S protein of SARS-CoV2 and a truncated version of human EGFR (huEGFRt) as reporter were used to detect anti-spike IgG1 in human sera as described in Horndler et al[31]. Briefly, for each serum to test 120 x $10^3$ cells were stained with a 1/50 serum dilution followed by a secondary anti-human IgG1 PE (clone HP6001, Southern Biotech) and an anti-human EGFR Bv421 (clone AY13, BioLegend) and the viability marker 7AAD. Samples were analyzed on a FACSCanto II flow cytometer and FlowJo software (Becton-Dickinson). The IgG1 PE/EGFR Bv421 ratio was used to determine the relative amount of anti-spike IgG1 signal.

## Neutralization assay with pseudotyped virus

Neutralization assays were performed as described in Horndler et al[31]. Briefly, lentiviruses were produced by co-transfecting plasmids pCMV-dR (gag/pol), pHRSIN-GFP and a truncated S envelope (pCR3.1-St). A total of $35 \times 10^3$ ACE2+ HEK293T cells per well in a 48-well plate were seeded the day before transduction. Serially diluted serum was incubated with viral supernatant for 1 h at 37˚C prior addition to the cells. Polybrene (8 μg/ml) was added and plates were centrifuged for 70 min at 1,600 g and left in culture for 48 h and then were resuspended in PBS with 2% FBS and 5 mM EDTA and fixed with 2% paraformaldehyde. GFP + cells were then analyzed on a FACSCanto II flow cytometer, and the data were analyzed with FlowJo software (Becton-Dickinson). Samples were run in duplicates and infection was normalized to 100% with viral supernatant without serum. A decrease in the neutralization capacity was observed as patient sera were serially diluted 1/50, 1/500 and 1/2000 (S6 Fig). The serum dilution 1/500 was chosen for comparison of neutralizing capacity between cohorts as it allowed the maximum discrimination. A known convalescent serum was used as positive control and a pre-pandemic serum was used as negative control.

## Viral load quantification

SARS-CoV-2 viral load at diagnosis was assessed using cycle threshold (Ct) values from a RT-PCR assay applied to nasopharyngeal swab samples. The RT-PCR assay was a laboratory-developed test (LDT) based on real time RT-PCR for detection of the E gene on the Panther Fusion Hologic (San Diego, CA, USA) using its open access functionality as previously described [53,54]. Amplification Ct values were considered a relative measure of viral load quantification. Lower Ct values corresponded with higher viral loads.

## Peripheral blood lymphocyte populations

EDTA-treated whole blood (50 μL) was stained with 20 μL of BD Multitest 6-color TBNK reagent in Trucount tubes (BD Biosciences. San Jose, CA, USA) for 15 min. After lysis of red blood cells, T, B and NK lymphocyte subsets were enumerated with a FACSCanto II flow cytometer, and data were analyzed using FACSDiva software (BD Biosciences) [55].

For the analysis of T helper cell subsets after vaccination, EDTA-treated whole blood was incubated with: anti-CD3-PerCPy5.5 and anti-CD4-KO (Beckman Coulter); anti-CXCR3-PE, anti-CCR6-PB and anti-PD-1-PECy7 (BD Biosciences). Acquisition was performed with a Navios flow cytometer (Beckman Coulter), and data were analyzed with FlowJo software v10.6.2. Th1 cells were defined as CD4+CXCR3+CCR6-, Th2 cells as CD4+CXCR3-CCR6- and Th17 cells as CD4+CXCR3-CCR6+. PD-1 was considered as a recent activation marker.

## Statistical analysis

Quantitative data were shown as the median with IQR, and qualitative variables were expressed as absolute and relative frequencies. Non-parametric Mann-Whitney U or Wilcoxon signed-rank tests were applied for comparison within two groups, when necessary. Kruskal-Wallis test was used to compare three or more unmatched groups. Correlations between continuous variables were evaluated using Spearman's rank test. Operating characteristics (ROC) curve analysis was performed to determine positivity cut-off values for cellular response. Univariate and multivariate logistic regressions were performed to test association with disease severity. Differences were considered statistically significant when $p < 0.05$. Statistical analysis was performed using GraphPad Prism version 8.0 software (GraphPad Software Inc, LaJolla, CA) and R software v4.0.3.

## Supporting information

**S1 Fig. Receiver operating characteristics (ROC) curve analysis and positivity cut-off values for S1, M and N cellular response.** Cut-off values were established by using a control group of 30 healthcare workers with no microbiological or clinical evidence of SARS-CoV-2 infection and 234 recovered COVID-19 patients. (**A**) S1 Cut-off >25 IFN-$\gamma$ sfu/$10^6$ PBMC (AUC = 0.97, 95%CI = 0.95–0.99). (**B**) M Cut-off >21 IFN-$\gamma$ sfu/$10^6$ PBMC (AUC = 0.97, 95% CI = 0.95–0.99). (**C**) N Cut-off >14 IFN-$\gamma$ sfu/$10^6$ PBMC (AUC = 0.96, 95%CI = 0.94–0.98). AUC: Area Under the Curve; IC: Confidence Interval.
(TIF)

**S2 Fig. Representative T cell IFN-$\gamma$ FluoroSpot images.** FluoroSpot IFN-$\gamma$ responses against SARS-CoV-2 S1, M and N peptide pools and positive (anti-CD3 and anti-CD28 mAb) and negative (anti-CD28 mAb only) control wells for mild (**A**), moderate (**B**) and severe (**C**) patients.
(TIF)

**S3 Fig. Development of SARS-CoV-2-specific cellular response in natural infection.** IL-2 and bifunctional IFN-$\gamma$+IL-2 T cell responses against SARS-CoV-2 S1, M and N protein pools according to the days post symptom onset during acute and convalescent phase in mild (green) (**A-B**) moderate (orange) (**C-D**) and severe (red) (**E-H**) patients. Data is represented as spot forming unit (sfu) per million PBMC. (**I-K**) Relative Spot Volume (RSV) per spot of secreted IL-2 after S1, M and N peptide pool stimulation in mild, moderate, and severe patients during acute infection. Horizontal bars and whiskers represent median values and interquartile ranges, respectively. The significance between groups was determined using Mann Whitney, Wilcoxon signed rank or Kruskal-Wallis tests, *$p < 0.05$, **$p < 0.01$, ***$p < 0.001$, ****$p < 0.0001$.
(TIF)

**S4 Fig. Analysis of the SARS-CoV-2- specific cellular response relative to the overall T cell functionality.** (**A**) FluoroSpot IFN-$\gamma$ responses against anti-CD3mAb polyclonal stimulus in mild (green), moderate (orange) and severe (red) COVID-19 patients. Data is represented as spot forming unit (sfu) per million PBMC. (**B-D**) S1, M and N cellular response adjusted by anti-CD3 sfu/$10^6$PBMC normalization according to the days post symptom onset during acute and convalescent phase in mild (green), moderate (orange) and severe (red) COVID-19 patients. Data is shown as relative sfu per million PBMC. (**E**) Relative SARS-CoV-2-specific IFN-$\gamma$-producing T-cell responses reactive to the S1, M and N proteins in mild, moderate, and severe patients during the first (W1) and the second (W2) week post symptom onset.

Horizontal bars and whiskers represent median values and interquartile ranges, respectively. The significance between groups was determined using Mann Whitney, Wilcoxon signed rank or Kruskal-Wallis tests, *p<0.05, **p<0.01, ***p<0.001, ****p<0.0001.
(TIF)

**S5 Fig. Hyperactivation state of T cell after natural infection.** (**A**) Comparison of Fluoro-Spot IFN-γ responses against anti-CD3mAb polyclonal stimulus in healthy controls (HC) and moderate or severe recovered patients (Post-COVID). (**B**) anti-CD3mAb IFN-γ responses according to the months post symptom onset (PSO). (**C**) S1, M and N cellular response normalized by anti-CD3 sfu/$10^6$PBMC, according to the months PSO. Data is shown as relative sfu per million PBMC. Horizontal bars and whiskers represent median values and interquartile ranges, respectively. The significance between groups was determined using Mann Whitney test, *p<0.05, **p<0.01, ***p<0.001, ****p<0.0001.
(TIF)

**S6 Fig. Neutralization capacity triggered after natural infection and BNT162b2 vaccination.** Neutralization capacity represented as normalized infection achieved in the presence of 1/50, 1/500 and 1/2000 (from left to right) serum dilutions for each sample in (**A**) Mild COVID-19 patients; (**B**) Moderate COVID-19 patients (Mod); (**C**) Severe COVID-19 patients (Sev); (**D**) Moderate or severe COVID-19 recovered patients (Post) and (**E**) after BNT162b2 vaccination (Vac). Coding sample in natural infection, 1: acute phase; 2: hospitalization; 3: convalescent phase. Coding sample in vaccination, 1': pre-boost; 2': 15 days after complete vaccination; 3': 30 days after complete vaccination. Horizontal bars and whiskers represent median values and interquartile ranges, respectively.
(TIF)

**S1 Table. Lymphocyte populations in successive phases of the disease according to severity.**
(DOCX)

## Acknowledgments

We would like to thank all patients, nurses, medical colleagues and healthy volunteers who contributed to the study.

## Author Contributions

**Conceptualization:** Patricia Almendro-Vázquez, Rocio Laguna-Goya, Jose Maria Aguado, Estela Paz-Artal.

**Formal analysis:** Patricia Almendro-Vázquez, Alberto Utrero-Rico, Pilar Delgado, Francisco Javier Gil-Etayo, Balbino Alarcon.

**Funding acquisition:** Jose Maria Aguado, Estela Paz-Artal.

**Investigation:** Patricia Almendro-Vázquez, Rocio Laguna-Goya, Alberto Utrero-Rico, Pilar Delgado, Miguel Moreno-Batanero, Marta Chivite-Lacaba, Francisco Javier Gil-Etayo, Carmen Martín-Higuera, María Ángeles Meléndez-Carmona, Irene Arellano, Luis Miguel Allende.

**Methodology:** Patricia Almendro-Vázquez, Rocio Laguna-Goya, Miguel Moreno-Batanero, Estela Paz-Artal.

**Project administration:** Jose Maria Aguado, Estela Paz-Artal.

**Resources:** Maria Ruiz-Ruigomez, Antonio Lalueza, Guillermo Maestro de la Calle, Pilar Delgado, Luis Perez-Ordoño, Eva Muro, Juan Vila, Isabel Zamarron, Carlos Lumbreras, Balbino Alarcon, Luis Miguel Allende, Jose Maria Aguado, Estela Paz-Artal.

**Supervision:** Rocio Laguna-Goya, Estela Paz-Artal.

**Visualization:** Patricia Almendro-Vázquez, Rocio Laguna-Goya, Alberto Utrero-Rico, Pilar Delgado.

**Writing – original draft:** Patricia Almendro-Vázquez, Rocio Laguna-Goya, Estela Paz-Artal.

**Writing – review & editing:** Patricia Almendro-Vázquez, Rocio Laguna-Goya, Maria Ruiz-Ruigomez, Alberto Utrero-Rico, Antonio Lalueza, Guillermo Maestro de la Calle, Pilar Delgado, Luis Perez-Ordoño, Eva Muro, Juan Vila, Isabel Zamarron, Miguel Moreno-Batanero, Marta Chivite-Lacaba, Francisco Javier Gil-Etayo, Carmen Martín-Higuera, María Ángeles Meléndez-Carmona, Carlos Lumbreras, Irene Arellano, Balbino Alarcon, Luis Miguel Allende, Jose Maria Aguado, Estela Paz-Artal.

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
