## [Decision Letter · Decision Letter 0]

20 Oct 2021

Dear Dr Laguna,

Thank you very much for submitting your manuscript "Longitudinal dynamics of SARS-CoV-2-specific cellular and humoral immunity after natural infection or BNT162b2 vaccination" for consideration at PLOS Pathogens. As with all papers reviewed by the journal, your manuscript was reviewed by members of the editorial board and by several independent reviewers. In light of the reviews (below this email), we would like to invite the resubmission of a significantly-revised version that takes into account the reviewers' comments.

Dear Dr. Laguna,

Thank you for submitting your manuscript. The reviewers appreciated your study and the need to validate findings in different cohorts. However, they had a major concern about the lack of novelty. If you are able to address this concern, perhaps by including additional data (such as the virologic parameters of infected patients, as suggested by Reviewer 1), or new immunologic results, we would be happy to reconsider this manuscript for publication.

We cannot make any decision about publication until we have seen the revised manuscript and your response to the reviewers' comments. Your revised manuscript is also likely to be sent to reviewers for further evaluation.

Sincerely,

Sujan Shresta

Associate Editor

PLOS Pathogens

Sonja Best

Section Editor

PLOS Pathogens

Kasturi Haldar

Editor-in-Chief

PLOS Pathogens

orcid.org/0000-0001-5065-158X

Michael Malim

Editor-in-Chief

PLOS Pathogens

orcid.org/0000-0002-7699-2064

Dear Dr. Laguna,

Thank you for submitting your manuscript. The reviewers appreciated your study and the need to validate findings in different cohorts. However, they had a major concern about the lack of novelty. If you are able to address this concern, perhaps by including additional data (such as the virologic parameters of infected patients, as suggested by Reviewer 1), or new immunologic results, we would be happy to reconsider this manuscript for publication.

Reviewer's Responses to Questions

**Part I - Summary**

Reviewer #1: This manuscript describes the kinetic of the development of humoral and cellular immune response in a cohort of 88 patients with mild, moderate and severe COVID-19 immediately after symptoms onset until recovery or fatal outcome. They also analyzed the SARS-CoV-2 specific immune response in a large group of convalescents and n some vaccinated individuals.

The work provides evidences that a development of a coordinated functional T and B cell response is associated with mild disease. Even though these findings are not novel ( see ref 15 and 16) , I agree with the authors that previous work was done in limited number of patients and as such this work constitutes an important confirmation of such data.

Reviewer #2: Almendro-Vázquez and colleagues investigated longitudinal kinetics of SARS-CoV-2-specific humoral and cellular immunity after either natural infection or BNT162b2 vaccination.

The authors analyzed S1-, M- and N-specific IFN-γ and IL-2 T cell immune responses and anti-S total and neutralizing antibodies in mild, moderate or severe acute COVID-19 patients.

They compared immune responses in COVID-19 patients with mild and severe disease. They also found a robust Th1-driven immune response in uninfected blood donors following BNT162b2-vacination. While understanding immune responses in COVID-19 is of a great importance, the novelty aspects of this study are unclear.

Specific comments:

The authors should specify upfront the novel aspect of their study in context of the literature.

Abstract: “Description of the immune response elicited by real-world anti-SARS-CoV-2 vaccination is still lacking”: there are numerous publications on immune responses following COVID-19 vaccine. This sentence should be rephrased.

Statistics on the graphs: there is no need to show 'ns' for not significant differences as these subtract from clear visualisation of significant differences.

Representative T cell Fluorospots appear to be too dark thus lack clarity.

**Part II – Major Issues: Key Experiments Required for Acceptance**

Reviewer #1: I have very little to say about the methods. Overall the data support the conclusions made.

A) However a weakness of the work is the lack of any virological quantification. It would be nice to add some virological parameters present in the studied patients. Quantity of virus present at the onset or a longitudinal analysis the persistence of SARS-CoV-2 +PCR over time in the different cohorts or in selected individuals might be a nice addition.

B) Figures quality is very poor. The impression i that the authors displayed all the data that they have but this doesn't increase the clarity of the message . The display of the results of Elispot are of very poor quality and they don't deliver any message, they are blacks dots identical in all the figures. It will be also nice to show some longitudinal data of selected patients in the different cohorts and make a direct comparison between mild, moderate and severe . The data are shown now as a sort of cross-sectional analysis an it is not clear whether some patients were analyzed sequentially. It will be more indicative and clear to show some of individual representative patients. Furthermore, there are 8 fatalities in their acute severe cohort according to their table, but they did not highlight them in their graphs. I think they should be highlighted, as it will be interesting to know if their immunological response is any difference from the other severe patients

Reviewer #2: (No Response)

**Part III – Minor Issues: Editorial and Data Presentation Modifications**

Reviewer #1: The T cell response to Spike was analysed only using peptides covering S1 regions. This should be discuss as a potential limitation of the analysis.

The authors should acknowledge in the results and not only in the introduction, the fact that their data confirmed previous works, I don't think the authors can state that" This is the first study analyzing the SARS-CoV-2-specific cellular and humoral immune response in patients upon ER arrival" . This was also done in ref 15-16

Reviewer #2: (No Response)

PLOS authors have the option to publish the peer review history of their article (what does this mean?). If published, this will include your full peer review and any attached files.

Reviewer #1: **Yes: **Antonio Bertoletti

Reviewer #2: No
---

## [Editor Report · Decision Letter 1]

2 Dec 2021

Dear Dr Laguna,

Thank you very much for submitting your manuscript "Longitudinal dynamics of SARS-CoV-2-specific cellular and humoral immunity after natural infection or BNT162b2 vaccination" for consideration at PLOS Pathogens. As with all papers reviewed by the journal, your manuscript was reviewed by members of the editorial board and by several independent reviewers. The reviewers appreciated the attention to an important topic. Based on the reviews, we are likely to accept this manuscript for publication, providing that you modify the manuscript according to the review recommendations.

Thank you for responding to the reviewers’ comments by adding new data and highlighting how your study is different from published ones. However, the discussion of the literature could be improved. In particular, several groups have now examined the durability of vaccine-induced immune responses beyond 3 months.

Sincerely,

Sujan Shresta

Associate Editor

PLOS Pathogens

Sonja Best

Section Editor

PLOS Pathogens

Kasturi Haldar

Editor-in-Chief

PLOS Pathogens

orcid.org/0000-0001-5065-158X

Michael Malim

Editor-in-Chief

PLOS Pathogens

orcid.org/0000-0002-7699-2064

Thank you for responding to the reviewers’ comments by adding new data and highlighting how your study is different from published ones. However, the discussion of the literature could be improved. In particular, several groups have now examined the durability of vaccine-induced immune responses beyond 3 months.

Reviewer Comments (if any, and for reference):

Figure Files:

Data Requirements:

Reproducibility:

References:

---

## [Editor Report · Decision Letter 2]

16 Dec 2021

Dear Dr Laguna,

We are pleased to inform you that your manuscript 'Longitudinal dynamics of SARS-CoV-2-specific cellular and humoral immunity after natural infection or BNT162b2 vaccination' has been provisionally accepted for publication in PLOS Pathogens.

Best regards,

Sujan Shresta

Associate Editor

PLOS Pathogens

Sonja Best

Section Editor

PLOS Pathogens

Kasturi Haldar

Editor-in-Chief

PLOS Pathogens

orcid.org/0000-0001-5065-158X

Michael Malim

Editor-in-Chief

PLOS Pathogens

orcid.org/0000-0002-7699-2064
---

## [Editor Report · Acceptance letter]

23 Dec 2021

Dear Dr Laguna,

We are delighted to inform you that your manuscript, "Longitudinal dynamics of SARS-CoV-2-specific cellular and humoral immunity after natural infection or BNT162b2 vaccination," has been formally accepted for publication in PLOS Pathogens.

Best regards,

Kasturi Haldar

Editor-in-Chief

PLOS Pathogens

orcid.org/0000-0001-5065-158X

Michael Malim

Editor-in-Chief

PLOS Pathogens

orcid.org/0000-0002-7699-2064